# Alternative to ZnO to establish balanced intestinal microbiota for weaning piglets

**Ákos Juhász**[1]*, **Viviána Molnár-Nagy**[2], **Zsófia Bata**[2], **Ko-Hua Tso**[2,3], **Zoltán Mayer**[1], **Katalin Posta**[1]

**1** Department of Microbiology and Applied Biotechnology, Institute of Genetics and Biotechnology, Hungarian University of Agriculture and Life Sciences, Gödöllő, Hungary, **2** Dr. Bata Ltd, Ócsa, Hungary, **3** Department of Animal Science, National Chung Hsing University, Taichung, Taiwan

* Juhasz.Akos@uni-mate.hu

**Data Availability Statement:** All relevant data are within the manuscript and its Supporting information files.

**Funding:** This research was supported by Development and Innovation Fund of Hungary,

## Abstract

A wide range of phytobiotic feed additives are available on the market claiming to have beneficial effects on the growth of the host animal and to promote the development of a balanced microflora. The present study investigated the effects of the phytobiotic-prebiotic mixture of curcumin, wheat germ, and chicory on the growth performance and on the intestinal microflora composition of weaning piglets. Post weaning diarrhea causes significant losses for the producers, most commonly it is prevented by feeding high doses of zinc oxide (ZnO). The effect of a phytobiotic-prebiotic feed additive (1 kg $T^{-1}$) was compared to a positive control (3.1 kg $T^{-1}$ ZnO) and to a negative control (no feed supplement) in an *in vivo* animal trial. There was no significant difference in the final body weight and average daily gain of the trial and positive control groups, and both groups showed significantly ($P<0.05$) better results than the negative control. The feed conversion ratio of the phytobiotic-prebiotic supplemented group was significantly improved ($P<0.05$) compared to both controls. Both phytobiotic-prebiotic mixture and ZnO were able to significantly reduce ($P<0.05$) the amount of coliforms after weaning, even though ZnO reduced the amount of coliforms more efficiently than the trial feed additive, it also reduced the amount of potentially beneficial bacteria. Metagenomic data also corroborated the above conclusion. In the trial and positive control groups, the relative abundance of *Enterobacteriaceae* decreased by 85 and 88% between 3 weeks and 6 weeks of age, while in the negative control group a slight increase occurred. *Lactobacillaceae* were more abundant in the trial group (29.98%) than in the positive (8.67%) or in the negative (22.45%) control groups at 6 weeks of age. In summary, this study demonstrated that a phytobiotic-prebiotic feed additive may be a real alternative to ZnO for the prevention of post weaning diarrhea and promote the development of a balanced gut system.

## Introduction

Weaning of piglets is a complex process [1], where changes of the gut microbiota may cause digestive disorders, primarily caused *Enterobacteriaceae* that include pathogens such as

grant number 2017-1.3.1-VKE-2017-00001 and by
Ministry of Innovation and Technology within the
framework of the Thematic Excellence Programme
2020, Institutional Excellence Subprogramme,
grant number TKP2020-IKA-12. The funders had
no role in study design, data collection and
analysis, decision to publish, or preparation of the
manuscript.

**Competing interests:** The authors have declared
that no competing interests exist.

*Salmonella enterica* [2, 3] and *Escherichia coli* (*E. coli*) [4–6]. For long time antibiotics or high
levels of zinc oxide (ZnO) have been used as feed additives for the prevention and treatment of
these post-weaning disorders in piglets [7]. Many studies proved that dietary supplementation
with high levels of ZnO effectively suppressed the growth of *E. coli* and other coliform bacteria
in weaning piglets [8–10]. However other studies showed adverse effects of high dietary ZnO,
such as reducing the numbers of intestinal beneficial bacteria, or increasing environmental
zinc emissions [11–13]. European legislation limits total dietary zinc to 150 mg kg$^{-1}$ in piglet
feed [14], and experts have already proposed to further reduce ZnO levels in swine diets [15].
As more and more countries are implementing restrictions on the use of medical ZnO levels
in swine diets alternatives ought to be investigated. Many *in vitro* and *in vivo* studies suggested
that phytobiotics and/or prebiotics could be alternatives to antibiotics and ZnO in animal diets
[16–18].

Phytobiotics are natural, biologically active substances that are abundant in essential oils
[19, 20]. *Curcuma longa* has long been used in traditional Asian culture, for its flavors and
health benefits. It's primary active substance is curcumin, a hydrophobic polyphenol [20]. *In
vitro* and *in vivo* studies showed that curcumin has various biological properties including
antioxidant, anti-inflammatory, and anti-carcinogenic effects [21–23]. Additionally, a variety
of gastrointestinal disease models demonstrated the protective effects of curcumin on the
intestinal mucosa barrier in humans and in animals [24–26]. These beneficial effects of curcu-
min fate it to be a favored ingredient in phytobiotic feed additives [20, 27].

Prebiotics, such as wheat germ and chicory, are non-digestible oligosaccharides that reach
the intestine undigested and stimulate the growth of beneficial bacteria [28]. Matteuzzi *et al.*
[29] showed that wheat germ has prebiotic effects due to its polysaccharide and raffinose con-
tent. These compounds resist digestion and reach the large intestine, where they affect the
colonic microflora by promoting the growth of *Bifidobacteria* and *Lactobacilli*. Chicory con-
tains large quantities of prebiotic compounds namely inulin and oligofructose [30]. These
compounds promote the growth of intestinal beneficial bacteria, induce the production of
pro- and anti-inflammatory cytokines, and reduce the diarrheal incidence in swine [30–32].

Post-weaning diarrhea is caused by disbalance of the intestinal microflora of piglets. ZnO
has long been used to control the proliferation of *E. coli* and reduce weaning diarrhea, however
legislation and ecological efforts require to reduce its use. Phyto- and prebiotic feed additives
may promote the development of a balanced microbiota of piglets and may prevent diarrhea.
Thus, the objective of this study was to evaluate the effect of such a feed additive on the intesti-
nal microbiota composition and on the growth performance in weaned piglets, to assess its use
as an alternative to ZnO.

## Materials and methods

### Animals, diets, and experimental design

The animal experiments were conducted in strict accordance with the guidelines of Hungarian
Government decree No. 40/2013. (II. 14.) and were approved by the Research Institute for
Animal Breeding, Nutrition and Meat Science (Herceghalom, Hungary, approval number:
27/3/2015). No piglets were killed or injured in the study, only fresh fecal samples were col-
lected from the floor of the pen.

A total of 110 piglets (Hungarian Large White × Hungarian Landrace) ×
(Pietrain × Duroc), weaned at 28 ± 1 day of age were involved in two consecutive
experiments in 2018 (E1; from 11 July 2018 to 3 October 2018, 12 weeks) and in 2019 (E2;
from 7 October 2019 to 6 January 2020, 13 weeks). Both experiments were performed at Her-
ceghalom, Hungary. Animals received a two-phase basal diet: a pre-starter diet from 2 weeks

of age until 5 weeks of age; and a starter diet from 5 weeks of age until the end of the trial (S1 and S2 Tables). During the nursing period, piglets had access to standard pre-starter feed from the second week after birth. Feed supplementation (with the trial feed additive or ZnO) started at 3 weeks of age and continued until the end of the experiment (S1 Fig). No other feed additives or antibiotics were included in any of the diets. Each pen was equipped with a stainless-steel feeder and a nipple drinker with *ad libitum* access to feed and water throughout the experiment.

Experimental groups received different diets: negative control (NC)–basal diet; positive control (PC)–basal diet supplemented with 3.1 kg $T^{-1}$ ZnO; and trial diet (T)–basal diet supplemented with 1 kg $T^{-1}$ phytobiotic-prebiotic feed additive containing curcumin, wheat germ extract, and zinc-chelate of tartaric acid, spray dried on chicory pulp.

In the E1 experiment, 54 piglets were randomly allocated into two treatments (T and PC) with 5 replicates per treatment (4–6 piglets per replicates). In the E2 experiment, 56 piglets were randomly allocated into three treatments (T, PC and NC) with 4 replicates per treatment (4–6 piglets per replicate). In each pen, there was an equal number of male and female piglets. Piglets were weighed individually at weaning (day 28 ± 1 day) and at the end of the experimental period (12 or 13 weeks), and average daily gains (ADGs) were calculated. Feed intake was recorded on a pen basis daily, and average daily feed intake (ADFI) and feed conversion ratio (FCR) were calculated for all treatment groups. Piglets were observed daily for signs of diarrhea from the start of the application of the pre-starter diet (two weeks of age). The severity of diarrhea was assessed visually by using a fecal consistency scoring (0, normal; 1, soft feces; 2, mild diarrhea; 3, semi liquid diarrhea and 4, liquid diarrhea) as described by Jamalludeen *et al.* [33].

## Sample collection and fecal microbiota composition by enumeration

One fecal sample per pen was collected at three time points during the E1 and E2 experimental periods: one week before weaning (3 weeks of age; before the start of feed supplementation), two weeks after the weaning (6 weeks of age) and at the end of the feeding trial (12 weeks of age). The three sampling time points (3, 6, and 12 weeks of age) are referred to as 3W, 6W, and 12W. Although the duration of the experiment was one week longer in E2, the fecal samples were collected at the same age in both cases (S1 Fig). The samples were collected during the peak hours of defecation in sterile fecal containers (Biolab Inc., Budapest, Hungary) from the pen floor. The floor was cleaned before and after sampling to avoid contamination.

The fecal samples were transported on ice to the laboratory, and processed immediately. The samples were homogenized with a sterile plastic spoon and small portions (4–5 g) of each sample were stored at -80 ˚C until DNA extraction and metagenomic analysis. Serial 10 –fold dilutions were prepared from the remaining portion of each sample in 1% Trypton saline (1 g $L^{-1}$ of Trypton dissolved in 8.5 g $L^{-1}$ of NaCl solution). One hundred μL from the $10^3$ to $10^7$ serial dilutions were plated in duplicate on different culture media: De Man, Rogosa and Sharpe Agar (MRS), Eosin Methylene Blue Agar (EMB), Nutrient Agar and Columbia Blood Agar media were used for lactic acid bacteria (LAB), coliform-, total aerobic and anaerobic bacteria, respectively. All media were purchased from Biolab Inc. (Budapest, Hungary). The inoculated agar plates were incubated at 37 ± 1 ˚C for 24–48 hours using aerobic or microaerophilic/anaerobic conditions (the latter for LAB and anaerobic bacteria). Number of colonies (colony forming unit, CFU) on the Petri dishes were counted in the range of 10–300 CFU $plate^{-1}$, and the number of viable bacteria were calculated per gram of feces (CFU $g^{-1}$ feces, wet weight). The results are presented as $log_{10}$-transformed data.

## DNA extraction and metagenomic analysis

The metagenomic analysis involved all fecal samples from the three treatments (T, PC, and NC) and three sampling time points (3W, 6W, and 12W) of the E2 experiment. DNA was extracted from the fecal samples (50 ± 1 mg) using a Quick-DNA Fecal/Soil Microbe Micro-prep Kit (ZYMO Research, CA, USA), the extraction procedure followed the manufacturer's instructions. The yield and purity of the DNA extracts were quantified using an Implen Nano-photometer P300 (Implen GmbH, München, Germany). Purified DNA from four samples (by pens) per treatment per sampling time were pooled, and were used for sequence analysis. The abundance of the bacterial communities of the fecal samples were measured by high-through-put sequencing on an Illumina MiSeq platform by UD-GenoMed Ltd. (Debrecen, Hungary). Amplification of the V3-V4 region of the 16S rRNA gene was performed following the recommendations of the 16S Metagenomic Sequencing Library Preparation Guide (Illumina, San Diego, CA, USA) [34].

DNA was amplified by 25 PCR cycles starting from 12.5 ng DNA using the KAPA HiFi HotStart Ready Mix (KAPA Biosystems, Wilmington, MA, USA). Each PCR cycles consisted of denaturing at 95 ˚C for 30 sec, annealing at 55 ˚C for 30 sec and extension at 72 ˚C for 30 sec. Post-amplification quality control was performed by an Agilent Bioanalyzer (Agilent Technologies, Santa Clara, CA, USA). MagSi-NGS$^{Prep}$ Plus (Magtivio BV, Limburg, Nether-lands) magnetic beads were used to purify the amplicons. The Index PCR was run with the Nextera XT Index Kit (Illumina, San Diego, CA, USA) with 502, 503, 504, 701, 702, 703, 704, 705, and 706 index primers. KAPA HiFi Hot Start Ready Mix was used for the subsequent PCR reaction, with: 8 cycles of denaturing at 95 ˚C for 30 sec, annealing at 55 ˚C for 30 sec, and extension at 72 ˚C for 30 sec. Before the library quantification MagSi-NGS$^{Prep}$ Plus mag-netic beads were used again to clean the PCR products. For the library validation, 1 μL of the diluted final library was run on a Bioanalyzer DNA 100 chip on an Agilent Bioanalyzer. Next, each library was normalized, pooled, and loaded onto an Illumina MiSeq platform for 2x250 bp paired-end sequencing (Illumina, San Diego, CA, USA).

The 16S rRNA gene paired-end amplicon reads were processed using the Frogs pipeline [35]. Briefly, forward and reverse reads were filtered and merged using VSEARCH [36] with the parameters: min amplicon size: 44; max amplicon size: 550; mismatch rate: 0.15. Merged sequences were clustered using Swarm [37]. Chimeric sequences were removed using *remo-ve_chimera.py* of the Frogs pipeline. Taxonomic assignment was performed using BLAST [38] against SILVA_SSU_r132_March2018 database [39] for ribosomal small-subunit RNA.

## Statistical analysis

Statistical analysis of piglet growth performances and CFU values were performed with R Sta-tistical Software 4.0.4 [40]. Values were expressed as means ± standard deviation. For the sta-tistical evaluation of growth parameters (body weight, ADFI, ADG, and FCR) and fecal bacterial enumeration, pens were used as the experimental unit (n = 4-6/pen for the growth parameters; n = 1/pen for the fecal bacterial enumeration). Differences between treatments were determined by one-way analysis of variance (ANOVA), followed by Tukey's post-hoc test. In all tests, a *P*-value < 0.05 was considered to indicate statistical significance. The alpha diversity indices were calculated using functions in the vegan package of the R environment [41]. Multidimensional scaling analysis (MDS) was performed using the 'plotly' function of the R environment. Linear discriminant analysis (LDA) coupled with the effect size (LEfSe algorithm) was used to identify characteristic biomarkers for each age group based on the abundance values [42]. Z-scores were calculated to demonstrate the abundance of the taxo-nomic profiles in each sample with the formula of $z = (x - \mu)/\sigma$, where x is the abundance of

the taxonomic profiles in each sample, $\mu$ is the mean value of the abundances in all samples, and $\sigma$ is the standard deviation of the abundances. The heatmap of the results was created with the R package pheatmap [43].

## Results

### Growth performance of piglets

The effect of feeding the phytobiotic-prebiotic feed additive (1 kg $T^{-1}$) on the growth parameters of piglets was evaluated in two independent trials, and compared to a positive control group with 3.1 kg $T^{-1}$ ZnO and a negative control group without any treatment (Table 1). There were no statistically significant differences in the body weight, ADFI, and ADG between T and PC groups in the E1 trial. FCR of PC was statically significantly higher than of the T group ($P<0.05$). In the E2 trial, both T and PC groups had statistically significantly higher body weights than the NC group ($P<0.05$), but there was no significant difference between body weights of the T and PC groups. In the E2 trial, the ADFI of the PC group was significantly higher than that of the T group or the NC group ($P<0.05$). In contrast, both the PC and T groups showed statistically significantly higher ADG than the NC group. The FCR of the T group was statically significantly lower than that of the PC or NC group. Overall, the growth parameters in the two experiments indicated that the phytobiotic-prebiotic feed additive performed equally or better than the PC group, and both the T and PC group performed significantly better than the NC group. Furthermore, fecal consistency scores varied between 0 to 1 through both experiments (S3 Table), hence no diarrhea was observed in any of the groups. These results show that from the production perspective the phytobiotic-prebiotic feed additive presents an alternative to ZnO for enabling the smooth weaning of piglets. Next, we sought to assess the effect of different treatments on the microbiome of the animals.

### Fecal microbiota composition by enumeration

The microbiota composition was characterized by traditional cultivation-based methods from excreted feces (Table 2). The first fecal samples were collected during the nursing period, before the start of differential feeding (3W).

The amount of aerobic, coliform and anaerobic bacteria was indistinguishable in the trial and positive control groups in E1 at the first sampling, in contrast the amount of LAB was significantly lower ($P<0.01$) in the T group than in the PC group. The difference in the amount of LAB was also present in E2, but in this case, the T group was significantly higher ($P<0.01$)

**Table 1. Growth performance of piglets in the trial (T), positive control (PC), and negative control (NC) groups during the two experiments.**

| Item | E1_T | E1_PC | E2_T | E2_PC | E2_NC |
|---|---|---|---|---|---|
| Number of piglets | 28 | 26 | 20 | 16 | 20 |
| Weaning weight, kg | 7.70±1.32 | 7.23±1.48 | 7.23±1.26 | 6.52±1.06 | 6.58±0.94 |
| Final weight, kg | 27.29±5.92 | 27.29±9.27 | 29.44±5.53[B] | 29.81±4.72[B] | 25.51±4.30[A] |
| ADFI, g/day | 773±176 | 842±98 | 736±21[A] | 879±42[B] | 731±50[A] |
| ADG, g/day | 363±103 | 371±153 | 352±82[B] | 370±66[B] | 300±59[A] |
| FCR | 2.13±0.07[a] | 2.27±0.13[b] | 2.09±0.07[A] | 2.38±0.09[B] | 2.44±0.25[B] |

Statistical significance between the groups was determined separately for the two experiments, based on the data measured at the end of the experiments. The experiment ended at days 56 and 63 post weaning for E1 and E2, respectively.

[a-b] Means with the different lowercase letters differ significantly ($P<0.05$) during the E1 experiment.

[A-B] Means with different capitals differ significantly ($P<0.05$) during the E2 experiment.

**Table 2. Fecal microbiota composition of piglets at different ages and diets.**

| Samples | Treatments | | | | |
|---|---|---|---|---|---|
| | E1_T[1] | E1_PC[1] | E2_T[2] | E2_PC[2] | E2_NC[2] |
| | Total aerobic bacteria ($\log_{10}$ CFU $g^{-1}$) | | | | |
| 3W | $8.70\pm0.33^{B}$ | $8.59\pm0.45$ | $8.88\pm0.57^{B}$ | $9.00\pm0.15^{B}$ | $8.46\pm0.42^{AB}$ |
| 6W | $7.88\pm0.32^{A}$ | $8.05\pm0.41$ | $9.07\pm0.11^{B}$ | $8.98\pm0.11^{B}$ | $9.12\pm0.34^{B}$ |
| 12W | $7.75\pm0.29^{A}$ | $7.63\pm1.38$ | $7.86\pm0.31^{A}$ | $7.94\pm0.81^{A}$ | $8.16\pm0.40^{A}$ |
| | Total coliform bacteria ($\log_{10}$ CFU $g^{-1}$) | | | | |
| 3W | $8.15\pm0.07^{B}$ | $8.40\pm0.59^{B}$ | $8.80\pm0.53^{B}$ | $8.31\pm0.54^{C}$ | $8.01\pm0.62$ |
| 6W | $7.50\pm0.64^{B}$ | $7.63\pm0.82^{B}$ | $8.07\pm0.48^{AB}$ | $7.57\pm0.26^{B}$ | $7.85\pm0.36$ |
| 12W | $5.45\pm0.79^{A}$ | $5.82\pm0.79^{A}$ | $7.44\pm0.56^{Ab}$ | $6.64\pm0.22^{Aa}$ | $7.43\pm0.65^{b}$ |
| | Total lactic acid bacteria ($\log_{10}$ CFU $g^{-1}$) | | | | |
| 3W | $8.65\pm0.10^{Aa}$ | $8.96\pm0.08^{b}$ | $9.04\pm0.22^{Ab}$ | $8.17\pm0.33^{Aa}$ | $7.88\pm0.59^{Aa}$ |
| 6W | $9.55\pm0.26^{C}$ | $9.17\pm0.40$ | $9.42\pm0.10^{Bb}$ | $8.85\pm0.07^{ABa}$ | $9.27\pm0.19^{Bb}$ |
| 12W | $9.06\pm0.25^{B}$ | $9.15\pm0.26$ | $9.53\pm0.28^{B}$ | $9.16\pm0.37^{B}$ | $9.31\pm0.14^{B}$ |
| | Total anaerobic bacteria ($\log_{10}$ CFU $g^{-1}$) | | | | |
| 3W | $8.98\pm0.19^{A}$ | $8.99\pm0.20^{A}$ | $9.08\pm0.28^{Ab}$ | $9.09\pm0.18^{b}$ | $8.21\pm0.46^{Aa}$ |
| 6W | $9.51\pm0.25^{B}$ | $9.46\pm0.27^{B}$ | $9.49\pm0.18^{B}$ | $9.29\pm0.25$ | $9.33\pm0.16^{B}$ |
| 12W | $9.38\pm0.22^{AB}$ | $9.70\pm0.45^{B}$ | $9.59\pm0.21^{B}$ | $9.31\pm0.27$ | $9.40\pm0.33^{B}$ |

[1]For E1, the number of samples was n = 3 (3W) and n = 5 (6W and 12W).

[2]For E2, the number of samples was n = 3 (3W) and n = 4 (6W and 12W).

Absence of letters signify that there was no statistically significant difference between the results.

[a-b] Means within a row (different treatments) with different lowercase letters differ significantly ($P<0.05$). Statistical significance between the groups was determined separately for the two experiments.

[A-C] Means within a column (different sampling times) with different capitals differ significantly ($P<0.05$). Statistical significance between the groups was determined separately for the two experiments.

than the PC and NC groups. The initial count of anaerobic bacteria was around $1\times10^9$ CFU $g^{-1}$ feces, except for E2_NC, where it was significantly lower ($P<0.01$). In the E1 experiment, there was no significant difference in the fecal microbiota composition between the T and PC groups at 6 and 12 weeks of age (two weeks after weaning and at the end of weaning period). In the E2 experiment, the amount of LAB in the 6W samples and the amount of coliforms in the 12W samples were significantly ($P<0.05$) lower in the PC group than in the T and NC groups.

The age of the animals appeared to have a greater influence on the microbiota composition than the different treatments. The mean number of aerobic bacteria decreased significantly ($P<0.05$) from 3 to 12 weeks of age. The decrease was continuous in E1, but in E2 higher CFU values were observed at 6W than at 3W, although the increase was not significant. The number of anaerobic bacteria increased during the experiments: between 3 and 6 weeks of age a significant ($P<0.01$) increase was observed in E1_PC, E2_T, and E2_NC, in the other cases the increase was not significant. The CFU counts of 6W and 12W samples hardly differed and the final anaerobic counts were between $2.04 \times 10^9$ and $5.01 \times 10^9$ CFU $g^{-1}$ feces. The number of coliform bacteria was above $10^8$ CFU $g^{-1}$ at the first sampling (3W) and decreased continuously from 3W to 12W. The decrease in the amount of coliforms was significant ($P<0.01$) between the first and the last sampling in all groups except NC, where the decrease was not significant. In the E1 experiment, the decrease in the number of coliforms was greater than in E2. The change in the amount of LAB was similar to that of anaerobic bacteria. In general, the amount of LAB in the first samples was lower than in the 6W and 12W samples, the increase was significant in all groups except in E1_PC.

## Fecal microbiota composition by sequencing

16S rDNA sequencing of all feces samples from the E2 trial characterized the fecal bacterial composition of the piglets and assessed the effects of different feed supplements. Isolated nucleic acid from four individual fecal samples was pooled separately for treatments and sampling times before sequencing. Samples were named based on sampling times and treatments. As the first samples (3W) were taken before the treatments started (S1 Fig), the effects of the treatments were assessed by comparing the samples taken at 6 and 12 weeks.

Illumina MiSeq sequencing generated 2,914,984 sequences from the nine pooled samples, one for each sampling time (3W, 6W, and 12 W) and each treatment (T, PC, and NC). The paired-end sequences, with expected length, were chimera filtered and grouped into clusters. All clusters containing less than 0.005% of all sequences were removed from further analysis. Finally, 1,334,221 sequences, grouped into 721 Operational Taxonomic Units (OTUs) were kept and identified based on 97% species similarity. The number of sequences of the individual samples ranged from 101,818 to 180,941 and contained 569 to 672 OTUs per sample. The average OTU number of the 3W samples was significantly lower compared to those of the 6W and 12W samples (S4 Table). All nine samples shared 320 OTUs and all OTUs were present at least in two different samples. Although there was no sample specific OTU, several unique OTUs were identified when comparing the treatments at 3, 6, and 12 weeks or the age-dependent groups (S2 Fig).

## Taxonomic profiles

A total of 14 phyla, 19 classes, 30 orders, 62 families, 195 genera and 277 species were detected by sequencing. Firmicutes and Bacteroidetes were the two most abundant phyla in all samples with an average abundance of 75.47% and 17.13%, respectively (Fig 1). Proteobacteria was the third most abundant phylum in 8 samples, with an average abundance of 4.43%, but this was only the fourth most abundant phylum in NC_3W sample (1.35%). In this sample (NC_3W), the third most abundant phylum was Euryarchaeota (2.16%), which proved to be the fifth in all other samples (0.56% in average, but ranged from 0.02% to 1.44% in all other samples). The average abundance of all other identified phyla was less than 1%, except for Actinobacteria (1.46%). Five phyla (Cyanobacteria, Patescibacteria, Synergistetes, Verrucomicrobia and WSP-2) were represented only by one OTU, although these OTUs appeared in almost all samples. The phylogenetic classification of samples showed greater (mainly age-related) differences at lower taxonomy ranks. At family level, the five most abundant families and their average abundance were *Lactobacillaceae* (13.46%), *Ruminococcaceae* (12.33%), *Prevotellaceae* (11.71%), *Lachnospiraceae* (10.40%) and *Clostridia 1* (8.69%), respectively. The average relative abundance of top five genera was 13.46%, 8.50%, 6.71%, 4.52% and 2.48% for *Lactobacillus*, *Clostridium sensu stricto 1*, *Streptococcus*, *Prevotella 9* and *Romboutsia*, respectively.

## Bacterial diversity

Alpha diversity indices showed the largest differences by age (S5 Table). The number of observed species ($P<0.01$) and the Shannon index ($P<0.05$) were significantly lower at 3 weeks than at 6 and 12 weeks. Chao1 ($P<0.01$) and ACE ($P<0.001$) indices showed significantly higher diversity at 6 weeks than at 3 and 12 weeks, but the Simpson indices did not differ significantly. The effect of the treatments was evaluated based on the merged data of the 6W and 12W samples, as the 3W samples were collected before the start of the use of feed additives. The differentially abundant taxa between different sampling times were confirmed with beta-diversity analysis, at 3W higher bacterial diversity was present than at 6W and 12W (S3 and S4 Figs).

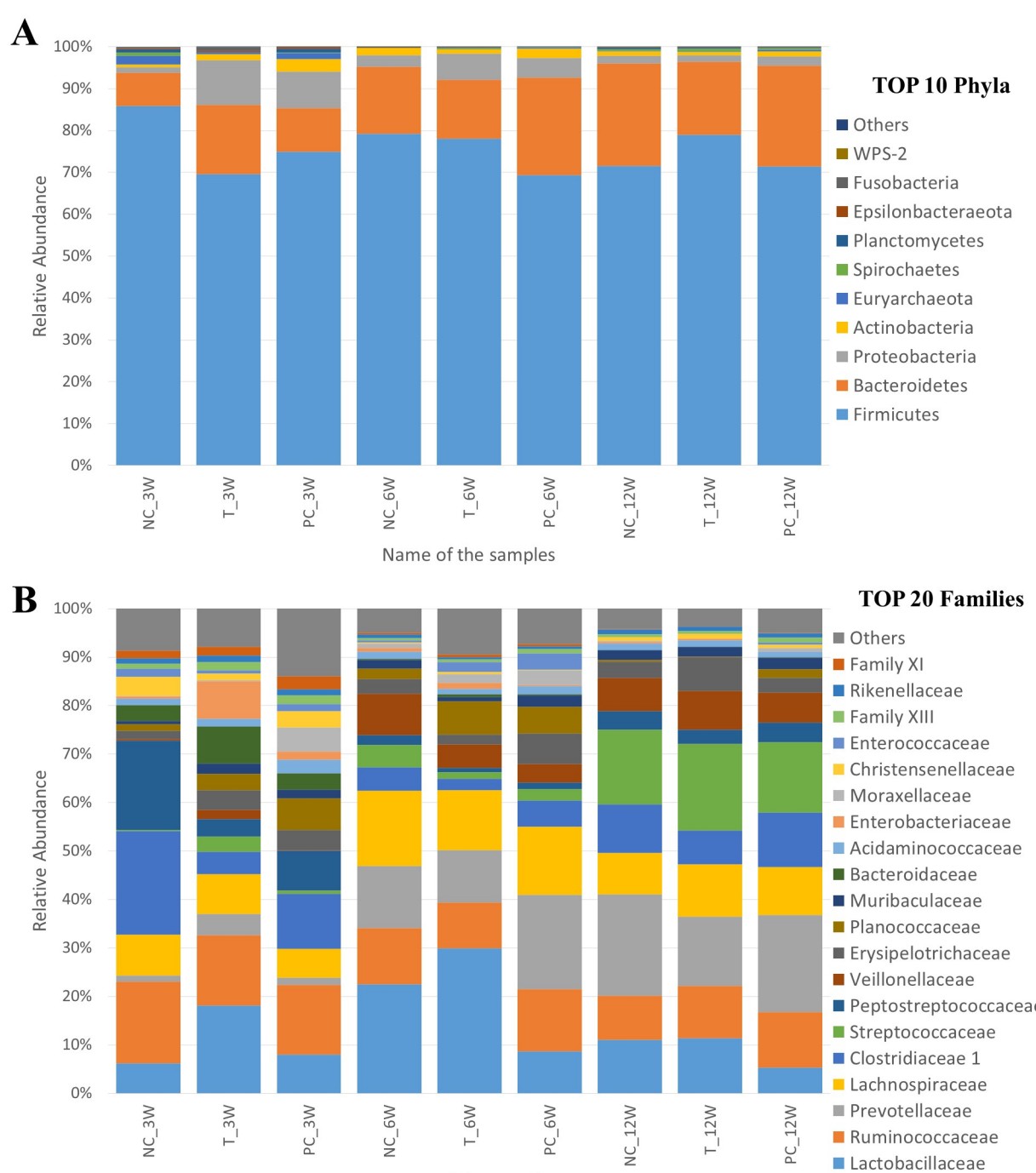

**Fig 1. Relative abundance of bacterial community of the nine fecal samples.** (A) Top 10 phyla. (B) Top 20 families.

## Age-related differences

The main age-related changes at phylum level was a decrease of Proteobacteria (from 6.91% to 1.86%), and an increase of Bacteroidetes (from 11.58% to 21.98%). Interestingly, the largest change occurred in Fusobacteria, where the average relative abundance decreased more than 400-fold, yet the relative abundance remained lower than 0.2% except for T_3W (0.97%).

The decrease of Enterobacteriales (3.37% to 0.22%) and Pseudomonadales (2.35% to 0.40%) induced the changes of Proteobacteria. Proteobacteria were represented by 38 OTUs in 3W and 6W samples but only by 31 OTUs in 12W samples (mainly due to the decrease of *Acinetobacter*-related OTUs).

Although the amount of Bacteroidetes increased steadily with age, certain taxonomic subgroups changed differently. Sphingobacteriales (1 OTU) were only observed in 3W samples, and Flavobacteriales were 21-fold more abundant in 6W samples than in 3W and 12W samples. Most of the families belonging to Bacteroidales decreased continuously with age, however the relative abundance of *Prevotellaceae* (85 OTUs) increased so significantly (from 2.35% to 18.42%), that the whole phylum increased. According to the linear discriminant effect size analysis (LEfSe) one of the 3W related feature was *Bacteroidaceae* and 12W related was *Prevotellaceae* (Fig 2). Most of the other significant features related to Clostridiales (341 OTUs) and younger age (3W: *Christensenellaceae*, *Family XI*, some *Ruminococcaceae* and *Romboutsia*; 6W: *Lachnospiraceae*). Although the average relative abundance of Clostridia decreased (from 52.22% to 34.93%), the abundance of other Firmicutes bacteria increased: Negativicutes (from 2.68% to 8.26%), Erysipelotrichia (from 3.27% to 4.43%), Bacilli (from 18.67% to 26.38%). In the case of Bacilli, the abundance of Bacillales decreased (from 3.82% to 0.77%) mainly due to the changes in *Planococcaceae* (*Lysinibacillus* was a significant 3W related feature). The relative abundance of Lactobacillales increased (3W: 14.85%, 6W: 27.02%, 12W: 25.61%), but the two most abundant families of Lactobacillales changed differently: *Lactobacillaceae* (17 OTUs) were more abundant in 6W (20.37%) than in 3W (10.77%) and 12W (9.25%), while the relative abundance of *Streptococcaceae* (3 OTUs) increased continuously (from 1.34% to 16.00%).

## Treatment-related differences

The microbial composition of treatments was compared at 6 and 12 weeks. At 6 weeks of age, the microbial composition of the NC and T samples (at all taxonomic levels) was more similar to each other than the PC sample (Fig 3). In the PC sample three phyla (Fusobacteria, Actinobacteria and Proteobacteria) and several families of Firmicutes, Bacteroidetes and Actinobacteria were more abundant. *Lactobacillaceae* were more abundant in T (29.98%) than in NC (22.45%) and in PC (8.67%). Clostridiales were less abundant in T (26.62%) than in NC (35.14%) and in PC (35.21%). *Enterobacteriaceae* and *E. coli* were also more abundant in T (1.20%) than in PC (0.21%), but in both cases it was lower than before the application of the feed additives (T_3W: 7.81%; PC_3W: 1.75%), thus in both T and PC groups the relative abundance of *Enterobacteriaceae* decreased by 85% and 88% between weeks 3 and 6. In contrast a slight increase was observed in the NC group (from 0.55% at 3W to 0.74% at 6W).

At 12 weeks of age, Firmicutes were approximately 7.5% more abundant in T than in the control samples (78.99% compared to 71.57% and 71.47%), while the relative abundances were lower in the next three most common phyla (Bacteroidetes, Proteobacteria and Actinobacteria) than in the control samples. *Lactobacillaceae* and *Streptococcaceae* were more abundant in T and in NC than in PC. Clostridiales were most abundant in PC (38.63%; NC: 32.90%, T: 33.25%). *Enterobacteriaceae* and *E. coli* were 0.32% and 0.31% in T and in NC, respectively, while they were only 0.04% in PC.

## Discussion

Many studies claim important effects of feed additives on the intestinal microbiota, the production and physiological parameters in piglets and *in vivo* microbiological and metagenomic studies are rarely presented together. In our study the piglet rearing data was analysed with the results of the traditional culture-based microbiological and modern metagenomic analyzes.

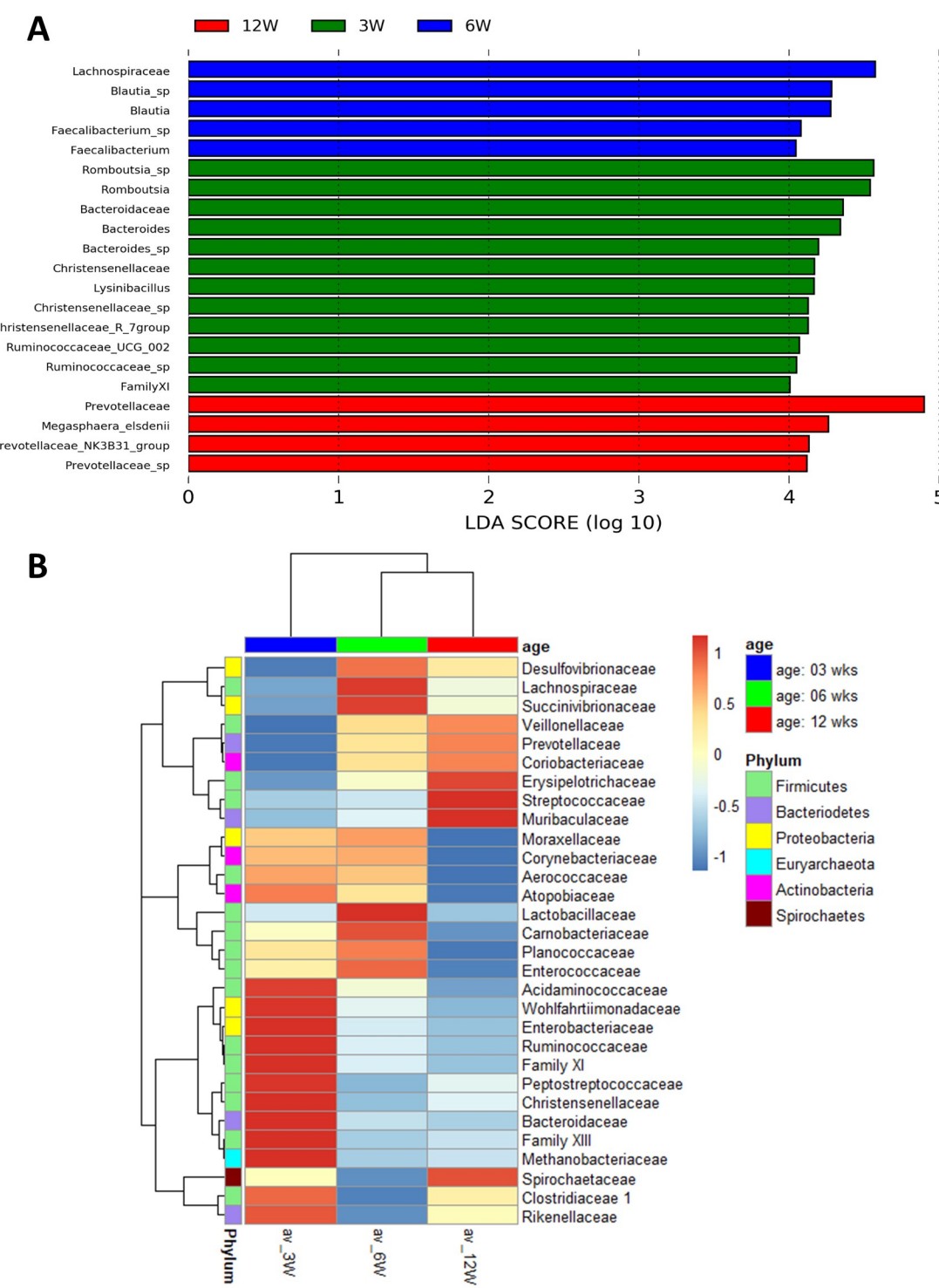

**Fig 2. Differentially abundant taxa among sampling times.** For the analysis of age dependent change the data of all treatments per age was merged. (**A**) Histogram of the results of LEfSe among 3, 6, and 12 weeks-age and their respective effect size; *P* values <0.05 considered significant. (**B**) Heatmap of the 30 most abundant families of bacteria at different ages. The color scale shows the Z-score of abundance of families within each group.

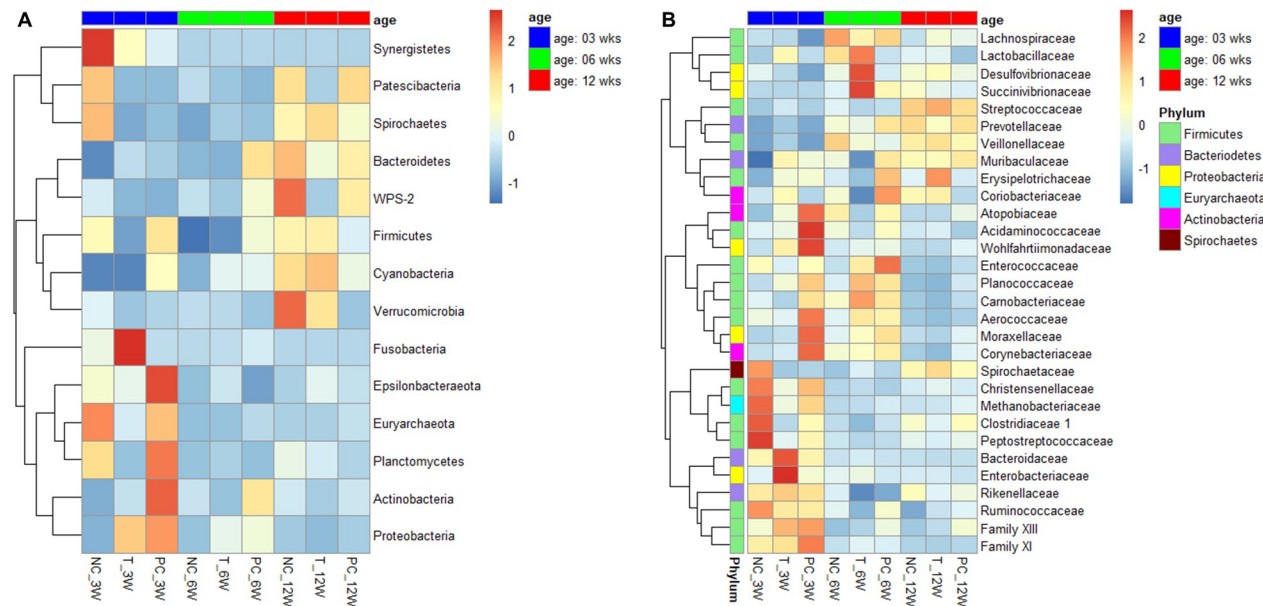

**Fig 3. Heatmap of (A) all identified phyla and (B) the 30 most abundant families of bacteria.** The color scale shows the Z-score of abundance of phyla and families within each group.

Changes due to the age and the general composition of the intestinal microbiota of the pigs have been thoroughly studied [44, 45]. and it was confirmed by our results, however we chose to focus primarily not on general but on treatment-dependent changes, with a particular focus on analyzing changes of microbiota caused by phytobiotic-prebiotic compound compared to the controls.

Our results show in two independent trials that there was no significant difference in the final body weight and ADG of the animals in the T groups receiving the feed additive in 1 kg T$^{-1}$ compared to the PC groups receiving ZnO in 3.100 kg T$^{-1}$ dosage. Furthermore, both groups showed a significantly ($P<0.05$) higher ADG than the NC group receiving no feed additive. The FCR value was significantly lower for the T groups in both experiments compared to the PC groups ($P<0.05$), consequently the T group receiving the feed supplement performed better than PC in terms of feed to weight conversion.

Similarly to our experiments, Xun *et al.* [20] reported that a high dose of curcumin (0.3 or 0.4 kg T$^{-1}$) significantly reduced the FCR compared to non-treated group, but the final weight and ADG were not affected. Shi *et al.* [46] also noted that 0.3 kg T$^{-1}$ curcumin and the combination of curcumin and piperine, in addition to several other beneficial effects, significantly lowered the FCR compared to non-treated group. Fermented wheat germ was reported to significantly enhance immune status and increase ADG [47]. No significant effects were reported for chicory for ADG, ADFI, and FCR compared to basal diet, although chicory did have a beneficial prebiotic effect [48, 49]. In our case, similar results were observed (FCR decreased and ADG increased only compared to the basal diet), as the additives were not used separately but together.

The beneficial and harmful effects of ZnO have been reported in numerous studies [50–52]. One of the best-known applications of ZnO in pig rearing is the prevention of post weaning diarrhea caused by *E. coli* [4, 5, 50], despite the fact that the antimicrobial effect of ZnO is not specific and has remarkable effects on porcine microbial populations [53]. For example, it can lower the amount of LAB [54, 55] and its antimicrobial effect is mainly addressed against

Gram-positive rather than Gram-negative bacteria [56]. Our CFU counts confirmed both observations: ZnO significantly reduced the number of coliforms in both experiments (E1, $P<0.001$; E2, $P<0.05$), and the amount of LAB was also lower at 6 weeks in the ZnO fed PC group than in the T and NC groups (significantly in E2, $P<0.05$).

In the E1 experiment, PC and T treatments performed similarly in reducing the number of coliform bacteria as the average CFU counts were similar. In the E2 experiment, the feed additives performed slightly differently, even though after weaning (6W) the difference in the amount of coliforms was not significant ($P>0.1$), it became significantly higher in the T group compared to the PC group at 12 weeks ($P<0.001$). In conclusion the data indicate that the phytobiotic-prebiotic feed additive adequately controlled the number of coliform bacteria. The amount of LAB was significantly lower in the PC group than in the NC, while the CFU counts of LAB of the T group were tendentiously or significantly higher than in the PC and NC groups. In agreement with our results, ZnO has been reported to reduce LAB in numerous studies [50, 54, 55], while the active ingredients of the phytobiotic-prebiotic feed additive (curcumin, wheat germ and chicory) are known to promote the growth of LAB [48, 57].

In order to better understand the effect of the phytobiotic-prebiotic feed additive on the gut microbial community, the culture based broad microbial counts were complemented with the highly detailed metagenomic analysis of the fecal samples. The metagenomic analysis was performed in the E2 trial, in which the effect of the trial feed supplement on the intestinal microbiota community was compared with the effect of negative and positive controls. The same fecal samples were analyzed in both culture dependent and independent methods, but before the metagenomic analysis, the isolated DNA samples were pooled, therefore one sample represents the average data of the feces of four piglets. The effect of the treatments was evaluated based on the data of the 6W and 12W samples.

Based on alpha and beta diversity analysis, the microbial community composition varied mainly with age and not treatment. The 3W samples differed more by groups than the 6W or 12W samples, of the 721 identified OTUs, the 3W samples shared only 464, while the 6W and 12W shared 587 and 574 OTUs, respectively. The greater diversity of 3W samples may originate from dissimilar early life colonization [58], where the gut microbiome is thought to be shared at least partially from the parents, and it becomes relatively stable once the dense microbial population is established later in life [59]. Although the microbial communities within one age group are less different in 6W and 12 W, the Chao1, ACE and Shannon indices show the diversity to be significantly higher ($P<0.05$) than at 3W. The largest alpha diversity was observed at 6W when the ACE index was significantly higher ($P<0.001$) than at 3W or 12W, while Chao1 and Shannon indices representing the observed species were tendentiously higher than at 12W. Previous studies also found that the alpha diversity increased with age at the early life of piglets [44, 45].

In accordance with previous reports, the two most abundant phyla were Firmicutes and Bacteroidetes in all samples. Proteobacteria were the third most abundant except in one of the 3W samples, which is particularly interesting as at this age Proteobacteria is expected to be one of the dominant phyla. In this sample the relative abundance of Proteobacteria was 1.35% while in the other two 3W samples it was 10.70 and 8.68%, respectively. This may be due to different parental and environmental factors which significantly affect the pig microbiome, as it was reported previously [49, 59].

The trends observed in the culture- dependent microbiota analysis were confirmed by metagenomic data for *Lactobacillaceae* and *Enterobacteriaceae*. In agreement with the CFU changes during the E2 experiment, lower *Enterobacteriaceae* values were obtained for the PC group than for the T group (6W: 0.21% vs. 1.20%; 12W: 0.04% vs. 0.32%), and the relative abundance of the beneficial *Lactobacillaceae* was also lower at both ages (6W: 8.69% vs.

29.98%; 12W: 5.30% vs. 11.39%). *Streptococcaceae* showed less differences than *Lactobacillaceae* with higher relative abundance in the T group (17.92%) than the PC group (14.60%) at 12W. Like *Lactobacillaceae*, members of *Streptococcaceae* are known for their beneficial effects and has been generally considered to be health-promoting microbes [60]. In our case the 3rd most abundant OTU of all samples was *Streptococcus gallolyticus* which was reported previously to be an important candidate for improving porcine feed efficiency [59]. The relative abundance of *Streptococcus gallolyticus* was also higher in the T group (17.67%) than in the PC group (14.29%) at 12W. Previous studies revealed *Christensenellaceae*, members of Clostridiales, to be associated with health and increased feed efficiency [61, 62]. In our case *Christensenellaceae* were more abundant in the T group than in the PC group (6W: 0.42% vs. 0.20%; 12W: 0.94% vs. 0.71%). These potentially beneficial bacteria were the most abundant in the T group and more abundant in the NC group than in the PC group in general, indicating that the phytobiotic-prebiotic feed additive increased the amount of beneficial microbes. The decrease of LAB and other beneficial microbes was reported several times for ZnO treatment [50, 54, 55]. The metagenomic analysis concluded that the relative abundance of coliforms in the T group was higher than in the PC group after weaning (at 6W), nonetheless the decrease of coliforms compared to pre-weaning (3W) was similar (T: 85%, PC: 88% reduction). The decrease in coliforms combined with an increase in beneficial bacteria in the T group successfully controlled the post-weaning diarrhea. The results also indicate that the general antimicrobial effect of ZnO can be replaced by a more specific effect of natural feed additives for preventing post-weaning diarrhea.

Interestingly, some other families (*Ruminococcaceae*, *Lachnospiraceae*, *Clostridiaceae 1*) belonging to Clostridiales were less abundant in the T group compared to the PC group at 6W, however the differences became only marginal at 12W with the exception of *Clostridiaceae* 1 which were 3–4% more abundant in the PC group than in the T group. *Prevotellaceae*, a family of Bacteroidales, showed larger differences and it was also more abundant in the PC group compared to the T group (6W: 19.47% vs. 10.80%; 12W: 20.09% vs. 14.25%). *Ruminococcaceae*, *Lachnospiraceae*, and *Prevotallaceae* as well as Clostridiales and Bacteroidales were reported to be associated with better feed efficiency and healthier microbiome [60, 63, 64] as these bacteria may enable a more efficient energy harvesting through the fermentation of various polysaccharides and dietary proteins [65, 66].

## Conclusion

This study compared the effects of curcumin, wheat germ, and chicory containing feed additive, ZnO and a basal diet on pig rearing and intestinal microbiome. The results demonstrate that the phytobiotic-prebiotic feed additive can be as effective as the commonly used ZnO in preventing post-weaning diarrhea and improve the feed efficiency of piglets. The feed additive was able to control the amount of potentially pathogenic bacteria and increase the number of health-promoting microbes (like *Lactobacillaceae*, *Streptococcaceae* and *Christensenellaceae*), unlike ZnO which have a non-specific general antimicrobial effect. The study used pharmaceutical levels of ZnO, 3100 mg kg$^{-1}$ (3.1 kg T$^{-1}$), while the current legislation only permits the use of 150 mg kg$^{-1}$ in regular piglet feed which unlikely exerts the same effects as pharmacological levels of ZnO. Therefore, we believe, that such a phytobiotic-prebiotic feed additive may be a sustainable alternative to ZnO in piglet rearing.

## Supporting information

**S1 Fig. Schematic design of the experiments.** The numbers (1–13) represent the average age of the animals in weeks. The dotted part represents the nursing period. The gray shaded part

indicates the continuous monitoring of feed consumption. The colored areas indicate the type of feeds and treatments in the T and PC groups.
(TIF)

**S2 Fig. Venn diagram of OTU distribution among different treatments. A**, **B** and **C**: differences between treatments at 3, 6, and 12 weeks. **D**: age-related differences.
(TIF)

**S3 Fig. Beta-diversity of the nine fecal samples.** (**A**) Heatmap highlighting the Jaccard diversity index among the microbial populations. The higher the color intensity, the lower the similarity between the pairs. (**B**) Multi-dimensional scaling of the data set. The dots of green, blue, and purple represent 3, 6, and 12 weeks, respectively.
(TIF)

**S4 Fig. OTU abundance based heatmap of all samples.**
(TIF)

**S1 Table. The composition and nutritional parameters of pre-starter feeds.**
(XLSX)

**S2 Table. The composition and nutritional parameters of starter feeds.**
(XLSX)

**S3 Table. Fecal consistency scores according to Jamalludeen** *et al.* [**33**] **in E2.**
(XLSX)

**S4 Table. Base sequence information of all samples.**
(XLSX)

**S5 Table. Alfa diversity indices of all samples and grouped by age or treatment.**
(XLSX)

## Acknowledgments

The authors wish to thank Beatrix Pethőné Rétháti for administrative support, Imréné Gódor and Csepregi Antalné for laboratory assistance and the staff of Research Institute for Animal Breeding, Nutrition and Meat Science for assistance in piglet rearing and sample collection.

## Author Contributions

**Conceptualization:** Ákos Juhász, Viviána Molnár-Nagy, Katalin Posta.

**Data curation:** Ákos Juhász, Zsófia Bata, Zoltán Mayer.

**Formal analysis:** Ákos Juhász, Viviána Molnár-Nagy, Zsófia Bata, Zoltán Mayer.

**Funding acquisition:** Viviána Molnár-Nagy, Katalin Posta.

**Investigation:** Ákos Juhász, Zsófia Bata, Zoltán Mayer.

**Methodology:** Ákos Juhász, Viviána Molnár-Nagy, Zsófia Bata, Zoltán Mayer, Katalin Posta.

**Project administration:** Ákos Juhász, Viviána Molnár-Nagy.

**Resources:** Katalin Posta.

**Supervision:** Ákos Juhász, Viviána Molnár-Nagy, Katalin Posta.

**Validation:** Ákos Juhász, Viviána Molnár-Nagy, Zoltán Mayer.

**Visualization:** Ákos Juhász, Zsófia Bata, Ko-Hua Tso, Zoltán Mayer.

**Writing – original draft:** Ákos Juhász, Viviána Molnár-Nagy, Ko-Hua Tso.

**Writing – review & editing:** Ákos Juhász, Viviána Molnár-Nagy, Zsófia Bata, Ko-Hua Tso, Katalin Posta.

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
