## [Decision Letter · Decision Letter 0]

21 Jan 2022

PONE-D-21-36256Alternative to ZnO to establish healthy intestinal microbiota for weaning pigletsPLOS ONE

Dear Dr. Ákos Juhász,

Thank you for submitting your manuscript to PLOS ONE. After careful consideration, we feel that it has merit but does not fully meet PLOS ONE’s publication criteria as it currently stands. Therefore, we invite you to submit a revised version of the manuscript that addresses the points raised during the review process. Please submit your revised manuscript by Mar 07 2022 11:59PM.  If you will need more time than this to complete your revisions, please reply to this message or contact the journal office at plosone@plos.org. Please include the following items when submitting your revised manuscript:A rebuttal letter that responds to each point raised by the academic editor and reviewer(s). You should upload this letter as a separate file labeled 'Response to Reviewers'.A marked-up copy of your manuscript that highlights changes made to the original version. You should upload this as a separate file labeled 'Revised Manuscript with Track Changes'.An unmarked version of your revised paper without tracked changes. You should upload this as a separate file labeled 'Manuscript'.

We look forward to receiving your revised manuscript.

Kind regards,

Saeed El-Ashram

Academic Editor

PLOS ONE

Journal Requirements:

Additional Editor Comments (if provided): Please accept my apologies for the delay

Reviewers' comments:

Reviewer's Responses to Questions

**Comments to the Author**

1. Is the manuscript technically sound, and do the data support the conclusions?

Reviewer #1: Yes

Reviewer #2: Yes

Reviewer #3: Partly

2. Has the statistical analysis been performed appropriately and rigorously? 

Reviewer #1: Yes

Reviewer #2: Yes

Reviewer #3: Yes

3. Have the authors made all data underlying the findings in their manuscript fully available?

Reviewer #1: Yes

Reviewer #2: Yes

Reviewer #3: Yes

4. Is the manuscript presented in an intelligible fashion and written in standard English?

Reviewer #1: No

Reviewer #2: Yes

Reviewer #3: Yes

5. Review Comments to the Author

Reviewer #1: Use the term “diarrhea” only for your recommendation for the future studies, since your experimental design does not contain diarrhea treatment.

For discussion part, try to give explanation why your results are similar to the literature results, such as mechanism or physiological effects on bacterial functions. For example, phenol compound affects cell wall of gram-negative bacteria. Or the effect of poly saccharides or your study bioactive compounds etc….

Reviewer #2: The paper could be accepted for publication after minor revision.

1. Please correct typos in your MS.

2. Please format the tables in your MS to meet Plos standard. or upload tables using a seperate file.

Reviewer #3: The possibility of using phytobiotic-prebiotic feed additive to replace zinc oxide in weaned piglets was studied in this ms, which was great significance in production. But There are some major questions I would like to bring up in order to make the ms better for publication.

Major issues

1.There are a number of grammatical errors and instances of badly worded/constructed sentences. Please ask someone familiar with English language to help you rewrite this paper.

2. The effect of calculating the average value of all samples in Table 3.

3. How is a healthy intestinal flora defined? According to the decrease of harmful bacteria and the increase of beneficial bacteria? In fact, intestinal flora is a complex environment, which can not be judged as healthy only by relying on several bacteria.

4. Zinc Oxide instead of antibiotics was used to alleviate diarrhea. Which indicator was used to measure diarrhea change after Phytobiotic-prebiotic feed additive adding?

Minor issues

1. Line 25-the letter “P”, indicated statistical difference, should be in italics along the text.

2. Line 29- “though” and “but” should not be used together.

3. Line 42, 63 and 187- “that” should not be used after the comma.

4. Line 200, 202 and 206- “feces” instead of “faces”.

5. Line 259- “fourteen” is written as “14” that can match the latter better.

6. PLOS authors have the option to publish the peer review history of their article (what does this mean?). If published, this will include your full peer review and any attached files.

Reviewer #1: No

Reviewer #2: No

Reviewer #3: No

---

## [Author Response · Author response to Decision Letter 0]

22 Feb 2022

Response to reviewers: 

Thank you for your comments and advice, we took it into focus when we have improved the manuscript. We respond to your review below:

General major changes:

- Language improvement: the language correctness of the manuscript has been checked.

- New result presentation: according to reviewer 1 and 3 we provided some new information about the monitoring of diarrhea. The fecal consistency scoring was assessed visually as described by Jamalludeen et al (2009). A new supplementary table was added too (S3 Tab). The original S3 Tab was renamed to S4 Tab (base sequence data).

- Table format change: the format of the tables in the manuscript has been changed to meet the requirements of PloS ONE. The original Table 3 was difficult to interpret (and understand) in this new format and was therefore transferred to the supporting materials section in its original format (S5 Tab). 

- Significant restructuring of the manuscript for better comprehensibility. Rewriting of the introduction, discussion and conclusion, and minor changes throughout the text.

Response to individual reviewers:

Reviewer 1:

1. Use the term “diarrhea” only for your recommendation for the future studies, since your experimental design does not contain diarrhea treatment.

The manuscript was expanded with a new table showing the fecal scoring system used to monitor diarrhea during the experiments (S3 Tab). The applied method is detailed in “Materials and Methods” (see line 105-109) while results are summarized in “Results” (see lines 199-201).

2. L65, add the hypothesis and L 66-68, I suggest you using this rephrased paragraph

The objective of this study was to evaluate the effect of phyto- and pre-biotic feed additives (curcumin, wheat germ and chicory) as alternatives to ZnO on intestinal microbiota composition and growth performance in weaned piglets. 

The introduction has been significantly redesigned and the aim of the study has been clarified. See lines L69-75, but the full introduction was reorganised or rewritten. 

3. L 71, provide a figure describing your experimental design.

The design of the experiment is described in S1 Fig.

4. L 229, delete “with different”

It was corrected. See line L241.

5. L 58-61, it is already stated at introduction section.

According to your suggestion we rewrote and shortened the introductory part of the discussion section (see lines 358-366) and changed some other parts of the discussion. Additionally, some part of the original discussion was moved to introduction section.

6. L 363, write it in more simple way

The introductory part of the discussion has been reworked, see above (5.). See lines L358-366.

7. L 354-369, it is too long introduction till you started to explain your results, I suggest you summarizing it.

The introductory part of the discussion has been shortened and reworked, see above (5.). See lines L358-366.

8. L 377, explain why your results are in agreement with the literature.

This part of the discussion was reworked too. See lines 374-383.

9. For discussion part, try to give explanation why your results are similar to the literature results, such as mechanism or physiological effects on bacterial functions. For example, phenol compound affects cell wall of gram-negative bacteria. Or the effect of poly saccharides or your study bioactive compounds etc….

In this study we investigated the in vivo effect of a phyto- and pre-biotic feed additive containing a mixture of curcumin, wheat germ and chicory. Curcumin, the main bioactive substance of our feed additive, is reported as a strong antioxidant, anti-inflammatory, antibacterial, antifungal, and antiviral agent. Both wheat germ and chicory have been described as prebiotic material. These studies were mentioned in the introduction and in the discussion sections, which were also significantly rewritten (see lines 54-60; 65-68; 374-381; 402-403 and related citations: 20-27; 30-32; 46-49; 57-58).

In our current work, individual active ingredients such as the flavonoid and polyphenol content of curcumin or the effect of prebiotic additives on microbes were not planned to be demonstrated. The main novelty of our work was the use of the combination of these compounds and in this study the detailed in vitro was not investigated, therefore we decided not to detail the hypothetical approach in this study and only to report what our own results support.

10. L 481, summarize the conclusion stating the most important results based on your measurements, in addition to the future application or invention which may be used based on your results.

The conclusion was reworked based on your suggestions. See lines 471-481.

Reviewer 2:

1. Please correct typos in your MS.

The language correctness of the manuscript has been improved and checked.

2. Please format the tables in your MS to meet Plos standard. or upload tables using a seperate file.

The format of the tables in the manuscript has been changed to meet the requirements of PloS ONE. The original Table 3 was difficult to interpret (and understand) in this new format and was therefore transferred to the supporting materials section in its original format (S5 Tab).

Reviewer 3:

Major issue 1.: There are a number of grammatical errors and instances of badly worded/constructed sentences. Please ask someone familiar with English language to help you rewrite this paper.

The language correctness of the manuscript has been improved and checked.

Major issue 2.: The effect of calculating the average value of all samples in Table 3.

At the request of Reviewer 2, we changed the format of the tables and finally decided to include Table 3 as a supporting material (S5 Tab). The content of the table and its accompanying text was not affected by this change. Average values were calculated by age and by treatments. For treatments, only data at 6 and at 12 weeks were considered, as treatments had not been started until 3 weeks of age. Although calculating the average value may not be the most appropriate method to compare individual groups, we did not have the opportunity to sequence multiple samples, so comparing averages was the only way to determine differences within treatments and within age.

Major issue 3.: How is a healthy intestinal flora defined? According to the decrease of harmful bacteria and the increase of beneficial bacteria? In fact, intestinal flora is a complex environment, which can not be judged as healthy only by relying on several bacteria. 

Based on your suggestion we renamed the term “healthy microbiota” to “balanced microbiota”. The intestinal microbiota is a complex community where not only microbes form relationships with each other, but it is also influenced in many ways by the host-intestinal microbiota interaction. Complete removal of (non-pathogenic) microbial groups, that are normally part of the intestinal microbiota, would presumably cause significant microbiota imbalance, and it is difficult to clearly describe some microbes as “beneficial” or “harmful”. 

We believe that the main purpose of the phytobiotic-prebiotic feed additive is to maintain the balance of the microbiota. During our experiments with the continuous feeding of the additive we found the increase of the number of certain microbial groups (like lactic acid bacteria), which are traditionally called “beneficial”, although there is no doubt that the number of other microbes has decreased in parallel. The increase in the number of beneficial microbes and the results of the production parameters obtained during pig rearing together confirmed that the effect of the applied feed additive was beneficial.

Major issue 4.: Zinc Oxide instead of antibiotics was used to alleviate diarrhea. Which indicator was used to measure diarrhea change after Phytobiotic-prebiotic feed additive adding?

According to your suggestion we provided some new information about the monitoring of diarrhea. The fecal consistency scoring was assessed visually as described by Jamalludeen et al (2009). A new supplementary table was added too (S3 Tab). Diarrhea was not observed in any of the experiments and fecal consistency scores remained in between 0 to 1 values. 

Minor issues

1. Line 25-the letter “P”, indicated statistical difference, should be in italics along the text.

2. Line 29- “though” and “but” should not be used together.

3. Line 42, 63 and 187- “that” should not be used after the comma.

4. Line 200, 202 and 206- “feces” instead of “faces”.

5. Line 259- “fourteen” is written as “14” that can match the latter better.

All listed issues have been corrected (or modified during grammatical revision).

---

## [Decision Letter · Decision Letter 1]

4 Mar 2022

Alternative to ZnO to establish balanced intestinal microbiota for weaning piglets

PONE-D-21-36256R1

Dear Dr. Juhász Ákos,

We’re pleased to inform you that your manuscript has been judged scientifically suitable for publication and will be formally accepted for publication once it meets all outstanding technical requirements.

Kind regards,

Saeed El-Ashram

Academic Editor

PLOS ONE

Reviewers' comments:

Reviewer's Responses to Questions

**Comments to the Author**

1. If the authors have adequately addressed your comments raised in a previous round of review and you feel that this manuscript is now acceptable for publication, you may indicate that here to bypass the “Comments to the Author” section, enter your conflict of interest statement in the “Confidential to Editor” section, and submit your "Accept" recommendation.

Reviewer #1: All comments have been addressed

Reviewer #2: All comments have been addressed

Reviewer #3: All comments have been addressed

2. Is the manuscript technically sound, and do the data support the conclusions?

Reviewer #1: Yes

Reviewer #2: Yes

Reviewer #3: Yes

3. Has the statistical analysis been performed appropriately and rigorously? 

Reviewer #1: Yes

Reviewer #2: Yes

Reviewer #3: Yes

4. Have the authors made all data underlying the findings in their manuscript fully available?

Reviewer #1: Yes

Reviewer #2: Yes

Reviewer #3: Yes

5. Is the manuscript presented in an intelligible fashion and written in standard English?

Reviewer #1: Yes

Reviewer #2: Yes

Reviewer #3: Yes

6. Review Comments to the Author

Reviewer #1: I suggest you these main lines in your conclusion section:

This study compared the effects of curcumin, wheat germ, and chicory feed additives on the growth performance and intestinal microflora composition of weaning piglets. Phytobiotic-prebiotic feed additives effectively prevented the post-weaning diarrhea compared to ZnO application. That potentially pathogenic bacteria abundance was decreased. The bioactive compounds extraction of these feed additives in form of therapeutic dietary powder may represent a sustainable management of piglets health in future.

Reviewer #2: (No Response)

Reviewer #3: Congratulations to all authors of this Manuscript. I think it is perfect enough to publish on PLOS ONE.

7. PLOS authors have the option to publish the peer review history of their article (what does this mean?). If published, this will include your full peer review and any attached files.

Reviewer #1: No

Reviewer #2: No

Reviewer #3: No

---

## [Editor Report · Acceptance letter]

9 Mar 2022

PONE-D-21-36256R1 

Alternative to ZnO to establish balanced intestinal microbiota for weaning piglets 

Dear Dr. Juhász:

I'm pleased to inform you that your manuscript has been deemed suitable for publication in PLOS ONE. Congratulations! Your manuscript is now with our production department. 

Kind regards, 

on behalf of

Professor Saeed El-Ashram 

Academic Editor

PLOS ONE